# Preparation of Citric Acid-Sewage Sludge Hydrochar and Its Adsorption Performance for Pb(II) in Aqueous Solution

**DOI:** 10.3390/polym14050968

**Published:** 2022-02-28

**Authors:** Yangpeng Huang, Dekui Shen, Zhanghong Wang

**Affiliations:** 1Key Laboratory of Energy Thermal Conversion and Control of Ministry of Education, Southeast University, Nanjing 210096, China; 220180401@seu.edu.cn; 2College of Eco-Environmental Engineering, Guizhou Minzu University, Guiyang 550025, China

**Keywords:** sewage sludge, hydrothermal carbonization, citric acid, Pb(II) adsorption

## Abstract

In order to seek the value-added utilization method of sewage sludge and develop low-cost and high-efficient adsorbents, a hydrochar was prepared by the co-hydrothermal carbonization of sewage sludge and citric acid and then characterized. The differences in Pb(II) adsorption performance between the citric acid–sewage sludge hydrochars (AHC) and the hydrochar prepared solely from sewage sludge (SSHC) were also investigated. When citric acid dose ratio (mass ratio of citric acid to dry sewage sludge) is 0.1, the obtained hydrohcar (AHC0.1) has the highest specific surface area (59.95 m^2^·g^−1^), the most abundant oxygen-containing functional groups, the lowest pH_pzc_ (5.43), and the highest equilibrium adsorption capacity for Pb(II). The maximum adsorption capacity of AHC0.1 for Pb(II) is 60.88 mg·g^−1^ (298 K), which is approximately 1.3 times that of SSHC. The potential mechanisms can be electrostatic attraction, co-precipitation, complexation, and cation-π interaction. It was demonstrated that by incorporating citric acid into the hydrothermal carbonization, resource utilization of sewage sludge can be accomplished effectively.

## 1. Introduction

Sewage sludge output in China rises in lockstep with the sewage treatment capacity. In 2020, the national sludge output reached 52.92 million tons [1]. Sewage sludge typically has a high moisture content, low density, poor dewaterability, and contains a considerable amount of harmful compounds such as organic pollutants, heavy metals. and pathogenic microorganisms [2,3], making its treatment and disposal troublesome under the stringent environmental regulations and relatively low subsidies [4]. As a result, the conventional sewage sludge treatment and disposal technologies are not effective and efficient enough to deal with the dilemma. Landfilling and soil utilization may result in the secondary pollution from organic contaminants and heavy metals, potentially exacerbating the already severe soil contamination in China [5]. Bio-chemical treatments, for example anaerobic digestion, have some drawbacks, such as the lengthy treatment cycle, limited treatment efficiency, and poor operational stability. Traditional thermal treatments such as incineration, pyrolysis, and gasification can recover the heat from sewage sludge, but the feedstock needs pre-drying. Recently, many novel drying methods, like roof solar drying methods [6] and chemical-assisted drying methods [7], have been developed, which can solve the problem of high energy consumption of the conventional thermal drying methods. However, the traditional thermal treatments are still not without drawbacks such as high complexity of the system, high construction cost, and the release of secondary environmental pollutants (NOx, SOx, dioxins, etc.) [8].

Recently, the hydrothermal carbonization has been considered to be a technology that can realize the reduction, harmless treatment, and resource utilization of sewage sludge effectively and efficiently. Hydrothermal carbonization of sewage sludge is typically conducted under 180~260 °C and spontaneous pressure (usually saturated pressure) without the need for pre-drying of its feedstock, resulting in relatively lower energy consumption compared with traditional thermal treatments [9,10,11]. Heavy metals could be enriched and stabilized in solid products during the hydrothermal carbonization of sewage sludge, and contaminants are released less, making it more ecologically benign [9].

The end products are mostly solid carbonaceous solids (hydrochar), organic-rich liquids (process water), and a little amount of gas (mainly CO_2_). The multiphase distribution of the products facilitates the comprehensive utilization of sewage sludge. The process water can be anaerobically digested for biogas production [12]. As for hydrochar, it has been widely studied and applied to energy recovery [13], water purification [5], and soil remediation [10,14]. Among them, the utilization of sewage sludge derived hydrochar as adsorbents for water remediation might be more competitive, profitable, and suitable for the volarization of sewage sludge.

Water pollution caused by heavy metals(especially Pb, Cr, Cd, etc.) has become a prominent challenge. Numerous treatment methods have been investigated and applied for heavy metal removal from wastewater, including adsorption, membrane methods (ultra-, nano-, micro-filtration, reverse osmosis, and electrodialysis), chemical method (precipitation, coagulation, and flocculation and flotation), ion exchange, electrochemical methods, photocatalytic methods, and coupled methods like adsorptional photocatalysis [15,16,17,18]. Compared with the others, adsorption has the advantages of easy operation, low cost, and high removal efficiency. Developing highly efficient and cost-effective adsorbents has attracted much attention and research interest, especially carbonaceous adsorbents prepared through thermo-chemical transformation of biowastes.

It has been reported that sewage sludge-based hydrochars (SSHC) are rich in surface functional groups and various minerals, which makes it suitable for the preparation of highly-efficient and low-cost adsorbents for heavy metal removal from water. The abundant oxygen-containing groups and nitrogen-containing groups in SSHC [19,20] (derived from the proteins, lignocelluloses, lipids, and other organics in sewage sludge) can coordinate or complex with heavy metal ions in the water. Heavy metal ions in aqueous solution can be also eliminated by precipitation with minerals like phosphates and silicates in SSHC. However, SSHC is of relatively poor pore structure, with a specific surface area of 6.3–17.3 m^2^·g^−1^ [19,20,21], which results in relatively few exposed adsorption sites and thus severely restricted adsorption performance. Thus, the additional activation was commonly adopted and the significant improvement was achieved [8,21], but the energy consumption greatly increased. In-situ modification seems more energy-saving and cost-effective.

Researchers have tried to incorporate additives into the hydrothermal carbonization of biomass and solid waste in order to increase carbonization degree, promote the etching of biomass, and or enhance ash removal. Hydrochloric acid, sulfuric acid, and phosphoric acid, as well as zinc chloride and aluminum chloride, are the most frequently used additives [22,23,24]. However, it is difficult to promote these technologies due to the corrosiveness of additives and the environmental issues associated with the use of inorganic acids and salts (highly salty and acidic wastewater discharge, etc.), even though the adsorption performance of the obtained hydrochars has been improved.

Citric acid is a readily available, environmentally friendly, and inexpensive organic acid that is frequently used as a cross-linking agent. As an additive into the hydrothermal carbonization of sewage sludge and other solid wastes, citric acid has been studied for improving the combustion performance of the products [25] and enhancing the recovery of phosphorus [26]. It has been found that citric acid, like the inorganic strong acids, can promote the hydrolysis and carbonization of biomass and the dissolution of ash during hydrothermal carbonization [25,27,28], as well as to enhance the cross-linking and stacking of carbon structures [29]. Its addition into the hydrothermal carbonization of sewage sludge can probably increase the specific surface area and surface functional groups of the produced hydrochar at the same time, which benefits its application as adsorbents.

Thus, in this study, the hydrochars were prepared through the co-hydrothermal carbonization of sewage sludge and citric acid and applied to eliminate Pb(II) from aqueous solution. The main objective of this study was to provide some insight into the effect of citric acid addition in the hydrothermal carbonization of sewage sludge on the adsorbents-related physicochemical properties and the heavy metal adsorption performance of the produced hydrochars. For this purpose, the specific tasks of this research were to: (1) characterize the produced hydrochars by elemental analysis, proximate analysis, diffuse reflectance Fourier transform infrared spectroscopy (DRIFT), X-ray photoelectron spectroscopy (XPS), pH_pzc_ titration, Boehm titration, X-ray diffraction spectroscopy (XRD), N_2_ adsorption/desorption isotherms, scanning electron microscopy (SEM), and thermogravimetric analysis(TGA); (2) investigate and compare the adsorption of Pb(II) onto the hydrochars by batch experiments; and (3) acquire the further insight into the potential adsorption mechanism.

## 2. Materials and Methods

### 2.1. Materials and Reagents

The sewage sludge (SS) was collected from a domestic wastewater treatment plant in Nanjing, with its water content of 83.4%. The chemical reagents used, including citric acid monohydrate (C_6_H_8_O_7_·H_2_O), lead nitrate (Pb(NO_3_)_2_), potassium nitrate(KNO_3_), nitric acid (HNO_3_), and sodium hydroxide(NaOH), were all analytically pure and purchased from Sinopharm chemical reagent Shanghai Co.,Ltd(Shanghai, China).

### 2.2. Preparation of SS-Derived Hydrochars

In the quartz liner of a 100-mL autoclave (TGYF-0.1L, Zhengzhou Taiyuan, Zhengzhou, China), 40 g of SS with an 85% water content was modulated, and citric acid (CA) was added at dose ratios (dose ratio (D_m_) = citric acid/SS dry weight) of 0.005 to 0.5. The autoclave was flushed with nitrogen gas (100 mL·min^−1^) for 10 min and then sealed with the inner gauge pressure maintaining 1.0 MPa. Then, the mixture was heated at 180 °C for 2 h. After allowing the reactor to cool naturally to the ambient temperature, the product was filtered and the filter residue was agitated for 24 h at 250 rpm in 500 mL of deionized water. The citric acid-sludge hydrothermal carbon (AHC) was obtained after drying at 105 °C for 8 h and then grinding through a 100 mesh screen, and the AHC prepared at a certain citric acid dose ratio (D_m_) was denoted as AHCD_m_, for example, when D_m_ = 0.1, the produced hydrochar can be referred to as AHC0.1. In the same manner as AHC, SSHC was produced solely from sewage sludge.

### 2.3. Characterization

The C, H, N, and S contents of the feedstock (dry SS) and the SS-derived hydrochars (including SSHC, AHC0.1, and AHC0.5) were determined using an elemental analyzer (Vario EL III, Elementar, Langenselbold, Hesse, Germany) and O content was calculated by difference, and the proximate analysis was carried out according to GB/T 28731-2012 “Proximate analysis method for solid biomass fuels”. DRIFT analysis (Nicolet iS50, Thermo Scientific, Waltham, MA, USA) was performed on SS and the SS-derived hydrochars, the scanning range: 4000 to 650 cm^−1^, the vertical axis was Kubelka–Munk. The mineral composition of the produced hydrochars was determined using XRD (Ultima IV, Rigaku, Tokyo, Japan) with the following operational parameters: Cu-Kα radiation, 40 KV/40 mA, 2θ = 5~90°, scanning speed = 2°/min. And X’pert Highscore Plus software (Malvern Panalytical) was used for XRD spectra analysis. The X-ray photoelectron spectroscopy (Nexsa, Thermo Scientific, Waltham, MA, USA) was applied to conduct the surface elemental analysis on the SS-derived hydrochars. The high-resolution spectra were calibrated by resetting the binding energy of C1s as 284.8 eV and de-convoluted using Avantage (Thermo Scientific). The surface area, pore volume, and pore size distribution of the hydrochars were characterized by Tristar II 3020 (Micromeritics Instruments, Norcross, GA, USA). For the hydrochars, pH_pzc_ was determined by the IT method described in Fiol’s study [30], and a Boehm titration was also performed [31]. Thermogravimetric analysis was conducted from 0 to 900 °C with a heating speed of 10 °C/min, under nitrogen atmosphere (N_2_ flow was 30 mL/min). Scanning electron microscopy was done using Regulus 8100 (Hitachi, Tokyo, Japan). The AHC0.1 after adsorption of Pb(II) (Pb@AHC0.1) was also characterized by XRD, DRIFT, and XPS for the adsorption mechanism study.

### 2.4. Adsorption Experiment

#### 2.4.1. Effects of Dm

The equilibrium adsorption experiments of Pb(II) by SS-derived hydrohcars prepared under different D_m_ (citric acid dose ratio in the AHC preparation) were conducted. For each experiment, 0.1 g of hydrochar was dosed into a 100 mg·L^−1^ Pb(II) solution of which pH had been adjusted to 4 using 0.1 M HNO_3_ solution and 0.1 M NaOH solution. The mixtures were shaken at 250 rpm and 298 K for 8 h in a thermostatic shaker.

#### 2.4.2. Effects of Adsorption Parameters

The adsorption of Pb(II) by SSHC, AHC0.1 and AHC0.5 was studied in terms of initial pH (1–8100 mg/L), reaction time (15–480 min), initial concentration of Pb(II) solution (100–500 mg·L^−1^) and temperature (25–55 °C). The volume of adsorbates and the dosage of hydrochars were set as 50 mL and 0.1 g, respectively, for all of the experiments After the adsorbed solution was filtered through a 0.45-μm membrane, the residual concentration of Pb(II) (C_t_, mg·L^−1^) was determined by ICP-OES, and the adsorbed amount and removal rate was hence calculated by Equations (1) and (2). Visual MINTEQ 3.0 was used to determine the Pb(II) ionic species distribution.

#### 2.4.3. Ion Exchange

SS-derived hydrochars contain a large amount of metal cations, mainly K^+^, Na^+^, Ca^2+^, Mg^2+^, Al^3+^, and Fe^3+^, their release into the solution in the adsorption process of Pb(II) can be caused by dissolution of salts, ion exchange with Pb(II) and H^+^. Hence, in order to assess the influence of phenomenological ion exchange on Pb(II) removal by AHC0.1, the effect of dissolution and H^+^ should be deducted. Thus, AHC0.1 was added at the same dosage ratio of 0.1 g/50 mL to the Pb(II) solution (solution α: 100 mg·L^−1^, pH was tuned by 0.1 M HNO_3_) and nitric acid solution at the same pH (solution β), shaken at 250 rpm and 298 K for 24 h. Then the difference of the change in the total amount of cation charge per unit mass of AHC0.1 (including H^+^, K^+^, Na^+^, Mg^2+^, Ca^2+^, Al^3+^, and Fe^3+^, but not Pb(II)) before and after agitation between solution α and solution β, which is named as Δ_qoc_, can be regarded as the result of ion exchange of them with Pb(II). The definition of Δ_qoc_ is presented in Equations (2) and (3). The concentration of the metal cationic ions and H^+^ in the solutions before and after agitation can be determined by ICP-OES and a pH meter, respectively.

### 2.5. Statistical Analysis

Adsorbed amount (q_t_, mg·g^−1^) was calculated by the following equation:(1)$$q_{t}=\frac{\left(C_{0}-C_{t}\right)\times V}{M}$$
where: C_0_ is the concentration of Pb(II) before adsorption, mg·L^−1^; V is the volume of adsorbate solution, L; M is the dosage of hydrothermal carbon, g.

The Equations (2) and (3) was used to calculate Δ_qoc_.
(2)$$\Delta_{\mathrm{qoc}}=\sum \delta_{i}I_{i}$$
(3)$$\delta_{i}=\frac{\left(C_{\mathrm{it}}^{\alpha }-C_{i0}^{\alpha }\right)}{M_{\alpha }}-\frac{\left(C_{\mathrm{it}}^{\beta }-C_{i0}^{\beta }\right)}{M_{\beta }}$$

And the definition of the ion exchange ratio (IER) is given as follow:(4)$$\mathrm{IER}=\frac{\Delta_{\mathrm{qoc}}}{2q_{e}^{\mathrm{Pb}}}$$
where: “i” represents cationic ions including H^+^, K^+^, Na^+^, Mg^2+^, Ca^2+^, Al^3+^, and Fe^3+^; I_i_ is the charge of the certain cationic ion; C_it_^α^ and C_it_^β^ are the concentration of the certain cation after agitation in solution α and solution β respectively, mmol/L; C_i0_^α^ and C_i0_^β^ are the concentration of the certain cation in solution α and β before agitation, mmol/L, and they are considered as 0 here. V_α_ and V_β_ are the volume of solution α and β, here V_α_ = V_β_ = 50 mL; M_α_ and M_β_ are the mass of AHC0.1 dosed into solution α and solution β, respectively, M_α_ = M_β_ = 0.1 g here; q_e_^Pb^ stands for the equilibrium adsorption amount of Pb, mmol/L. IER was used to assess the contribution of phenomenological ion exchange to the Pb(II) adsorption.

The pseudo-first-order and pseudo-second-order models were used for the adsorption kinetics fitting, and their non-linear forms are shown in Equations (5) and (6) [32].
(5)$$q_{t}=q_{e}-e^{\left(\ln\left(q_{e}\right)-K_{1}t\right)}$$
(6)$$q_{t}=\frac{K_{2}q_{e}^{2}t}{1+K_{2}q_{e}t}$$
where: q_e_ is the equilibrium adsorption uptake of adsorbate, mg·g^−1^; q_t_ is the adsorption uptake of adsorbate at time t, mg·g^−1^; K_1_ is the pseudo-first-order adsorption kinetic constant, and K_2_ is the pseudo-second-order adsorption kinetic constant. Non-linear regression was used to determine the coefficients.

Langmuir isotherm and Freundlich isotherm were used to simulate the adsorption isotherm of Pb(II) by SSHC and AHC0.1 in this study. Their linear forms are presented as follows [33]:(7)$$\ln\left(q_{e}\right)=\ln\left(K_{F}\right)+\mathrm{nln}\left(C_{e}\right)$$
(8)$$\frac{C_{e}}{q_{e}}=\frac{C_{e}}{q_{\max}}+\frac{1}{K_{L}}$$
where: q_e_ is the equilibrium adsorption capacity, mg·g^−1^; q_max_ represents the theoretical maximum adsorption capacity, mg·g^−1^. K_F_ stands for Freundlich constant, while K_L_ is Langmuir constant. The linear regression was applied for the parameter determination.

BET isotherm model, as a multilayer adsorption isotherm model, was also considered to simulate the data of Pb(II) adsorption by AHC0.1 and SSHC [34,35]:(9)$$\frac{C_{e}}{q_{e}\left(C_{s}-C_{e}\right)}=\frac{1}{q_{s}C_{\mathrm{BET}}}+\frac{\left(C_{\mathrm{BET}}-1\right)}{q_{s}C_{\mathrm{BET}}}\left(\frac{C_{e}}{C_{s}}\right)$$
where C_s_ stands for monolayer saturation concentration, mg·L^−1^, C_BET_ denotes BET constants, L·mg^−1^, q_s_ represents isotherm saturation capacity, mg·g^−1^.

Adsorption thermodynamic parameters include Gibbs free energy change (ΔG, J·mol^−1^), enthalpy change (ΔH, J·mol^−1^), and entropy change (ΔS, J·mol^−1^·L^−1^), and they follow the following equations:(10)ΔG=ΔH−TΔS
(11)$$\ln\left(K\right)=\frac{-\Delta H}{\mathrm{RT}}+\frac{\Delta S}{R}$$
(12)ΔG=−RTln(K)

Since it was proved that the adsorption of SSHC and AHC0.1 fitted Langmuir model well, then [36]:(13)$$K=K_{L}\frac{C_{r}}{\gamma_{e}}$$
where: T is the adsorption temperature, K; K is the equilibrium constant; R is the ideal gas constant, 8.314 J·mol^−1^·K^−1^; K_L_, L·mol^−1^; C_r_ is the reference concentration, which is regarded as 1 mol·L^−1^; γ_e_ is the ionic strength at adsorption equilibrium (the ion concentration is low, γ_e_ ≈ 1), that is, take the value: K = K_L._ The thermodynamic parameters can be obtained by plotting ln(K) against 1/T followed by linear fitting.

## 3. Results and Discussion

### 3.1. Characterization of SS-Derived Hydrochars

#### 3.1.1. Ultimate and Proximate Analysis

Ultimate analysis and proximate analysis of SS and the SS-derived hydrochars are listed in Table 1. The mass contents of C, H, O, N, and S in SS are 26.53%, 6.47%, 17.42%, 4.72%, and 0.72%, respectively, and drastically decrease by 26.69%, 50.08%, 41.85%, 66.53%, and 33.3%, individually, after the hydrothermal carbonization of SS, indicating the decomposition of massive organics from the solid phase into liquid and gas phase. SS has a H/C of 2.93 and an O/C of 0.49, respectively. The H/C and O/C values of SSHC (1.99 and 0.39, respectively) are significantly lower than those of SS, indicating that hydrothermal carbonization can enhance SS’s aromaticity and hydrophobicity. The proximate analysis reveals that hydrothermal carbonization could reduce the volatile content of SS (from 50.19% to 29.11%) but increase its fixed carbon content and ash content.

The addition of citric acid has an great influence on the elemental contents of AHC0.1 and AHC0.5 relative to SSHC, and this influence varies with the citric acid dose ratio. AHC0.5 (21.65%) has the largest C content among the hydrochars, while the C content of AHC0.1 (18.64%) is 4.16% lower than that of SSHC (19.45%). The H contents of AHC0.1 and AHC0.5 are 7.43% and 13.93% lower than SSHC, respectively. It is worth noting that the H/C value of AHC0.1 (1.92) and AHC0.5 (1.54) are both lower than that of SSHC (1.99), suggesting that the addition of citric acid can increase the carbonization degree of the products. The O/C value of SSHC, AHC0.1, and AHC0.5 are 0.39, 0.48, and 0.32, respectively, indicating that when D_m_ = 0.1, the hydrophilicity and the oxygen-containing functional groups quantity of the obtained hydrochar (AHC0.1) are higher than those of SSHC and AHC0.5. Additionally, it is shown in proximate analysis that AHC0.1 and AHC0.5 have lower ash contents than SSHC, suggesting that citric acid might actually etch the ash content.

#### 3.1.2. DRIFT and Boehm Titration

Figure 1 illustrates the DRIFT spectra of SS and the SS-derived hydrochars. The broad and blunt absorption peak near 3274 cm^−1^ is attributed to the stretching vibration of −OH and −NH [37,38]. The absorption peak between 3000 to 2800 cm^−1^ is related to the −CH3 and −CH_2_− groups in the saturated hydrocarbon chain. The peak at 1668 cm^−1^ [39] and 1645 cm^−1^ are caused by the stretching vibration of C=O in the carboxylic acid or amide I bands [40,41]. The absorption peaks at 1539 and 1558 cm^−1^ indicate the bending vibration of N−H in the amide II band (protein source) [41]. The absorption peaks at 1608 and 1454 cm^−1^ are the C=C stretching vibration peaks in the aromatic ring [40,42]; The peak located at 1157 cm^−1^ corresponds to glycoside C−O−C [40,42]. The absorption peak at 1057 cm^−1^ is related to C−O−R (ether) or Si−O [42]. The peaks at 810 and 698 cm^−1^ can be ascribed to the out-of-plane CH bending vibration peak or the vibration absorption peak of N−H in pyrrole and pyridine [42].

As the citric acid dose ratio increases, the peak intensity of the amide II band (1539 cm^−1^ and 1558 cm^−1^) reduces dramatically, whereas the absorption peak intensities of −NH/−OH (3274 cm^−1^) and saturated hydrocarbon chains (3000–2800 cm^−1^) decline. The intensities of the aromatic ring and nitrogen-containing heterocyclic ring (1608, 1454, 810, and 698 cm^−1^) slightly increase with citric acid dosage, indicating that the addition of citric acid might enhance the hydrolysis of proteins and promote the dehydration, deamination, and aromatization, thereby enhancing the aromaticity and carbonization degree of the produced hydrochar, which complies with the elemental analysis results.

In order to quantitatively analyze the influence of citric acid addition on the surface functional groups on the produced hydrochars, the Boehm titration method was employed. The results are shown in Table 2. In comparison with SSHC, AHC0.1 contains slightly more phenolic hydroxyl groups and has significantly higher content of lactone and carboxyl groups, and in the end its total amount of surface O-containing functional groups (1550 ueq·g^−1^) is also significantly higher. Additionally, the total amount of O-containing functional groups on the surface of AHC0.5 (660 ueq·g^−1^) is lower than that of SSHC. The results of Boehm titration are consistent with the interpretation of the elemental analysis. The pH_pzc_ of AHC0.1 is 5.34 (seen in Table 2), which is lower than that of SSHC and AHC0.5, confirming that the number of O-containing functional groups on the surface is strongly related to pH_pzc_ [43].

#### 3.1.3. XPS Analysis

The deconvolution results of C1s XPS spectra of SS-derived hydrochars are presented in Figure 2, in which C1, C2, C3, and C4 represent C−C, C−O/C−N, C=O, and O−C=O, respectively [14,44], and their relative contents are summarized in Table 3. The relative content of O−C=O on the surface AHC0.1 (9.56%) is significantly higher than that of AHC0.5 (6.22%) and SSHC (5.57%), which complies with the sequence of the content of lactone and carboxyl groups determined by Boehm titration. AHC0.5’s relative content of C−C (82.03%) is much higher than that of AHC0.1 (76.55%) and SSHC (74.91%), while its relative content of C−N/C−O is significantly lower than that of AHC0.1 and SSHC, implying the deamination and dehydration might be drastically enhanced in the preparation of AHC0.5.

The N1s fine spectra of SSHC, AHC0.1, and AHC0.5 are deconvoluted as N1(pyridine-N), N2(amine-N), N3(protein-N), N4(pyrrole-N), N5(quaternary ammonium-N), and N6(N=O) [13,43] and presented in Figure 2e–g and Table 3. With the increase of citric acid dosage, the relative contents of protein-N and N=O decrease, while the relative contents of pyrrole-N and quaternary ammonium-N change inversely, indicating that citric acid boosts the hydrolysis of protein and the formation of nitrogen-containing heterocycles, which agrees with the weakening amide band II in DRIFT’s findings.

#### 3.1.4. SEM

SEM was applied to study the morphology evolution from sewage sludge to hydrochars. As depicted in Figure 3a, dry sewage sludge can be regarded as clustered aggregates with smooth surface and few pores. Compared with dry sewage sludge, SSHC exhibits a much rougher surface with more pores, which might be caused by the release of some volatile contents. This also indicates that hydrothermal carbonization can develop pore structure. In comparison with SSHC, AHC0.1 and AHC0.5 display more fragments, flakes, and sphere-like microparticles adhered to them. It can be deduced that addition of citric acid can promote the degradation of macro-organics and increase the carbonization degree. However, the effect of citric acid addition on the pore characteristics need quantitative analysis through N_2_ adsorption/desorption isotherms.

#### 3.1.5. Nitrogen Adsorption/Desorption

Specific surface area (SSA), total pore volume and average pore size of SS-derived hydrochars are summarized in Table 4, and the N_2_ adsorption/desorption isotherm plots and the pore size distribution curves are shown in Figure 4. The N_2_ adsorption/desorption isotherm plots can be classified as type IV isotherm with H3 hysteresis loop [45], which indicates slit mesopores. The specific surface area of AHC0.1 is 59.95 m^2^·g^−1^, which is the largest among the SS-derived hydrochars and 31.67% higher than that of SSHC (45.53 m^2^·g^−1^), but the total pore volume values of AHC0.1 and SSHC are almost the same. The pores sizes of SSHC and AHC0.1 mainly fall within the range from 2 to 50 nm (seen in Figure 4), making them typically mesoporous materials. The average pore size and peak pore size of AHC0.1 are smaller than SSHC, which accounts for its higher value of specific surface area, implying citric acid can promote the formation of small pores at its dose ratio of 0.1. As for AHC0.5, its specific surface area and total pore volume are the poorest, the reason behind which might be the blockage of pores by macromolecular hydrophobic organics (like humic substances or Maillard products) generated.

#### 3.1.6. XRD Analysis

The XRD spectra of SSHC, AHC0.1, and AHC0.5 are shown in Figure 5a–c. They contain basically the same minerals, mainly quartz, aluminum phosphate (AlPO_4_), muscovites ((Na_0.98_Ca_0.02_) (Al_1.02_Si_2.98_O_8_), Na_1.96_Ca_0.04_Si_5.96_Al_2.04_O_16.00_ and Na_0.986_ (Al_1.005_Si_2.995_O_8_)), feldspars ((K_0.94_Na_0.06_) (AlSi_3_O_8_) and K_0.94_Na_0.06_Al_0.95_Si_3.05_O_8_), etc.. Although based on the XRD characterization results, the incorporation of citric acid had no obvious effect on the mineral components of the SS-derived hydrochar, but can effectively reduce its ash contents according to the proximate analysis. Pb(II) could be removed during the adsorption process through co-precipitation with these phosphates and silicates in these SS-derived hydrochars.

#### 3.1.7. TG/DTG

TG under N_2_ atmosphere was conducted and DTG curves were also considered for thermal properties of the hydrochars. The DTG curves can be divided as four weight loss regions. The first interval (region I) ranges from 0 to 200 °C and can typically be ascribed to the release of free water, bound water, and low-boiling-point organic compounds. Mass loss in the temperature interval of 200–400 °C(region II) can be attributed to the degradation of susceptible matters, including organic acids, alkyl moiety, and carbohydrates [46]. Region III ranges from 400 to 600 °C, of which weight loss resulted from the decomposition of aromatic compounds. And the transformation of inorganics occurred in the temperature interval of 600–800 °C (region IV). Apparently, the peak temperature in the DTG curve of SS is much lower than that of SSHC, with its DTG peak intensity much higher than that of hydrochars, indicating that hydrothermal carbonization can improve thermal stability. As seen in Figure 6b, the intensity of DTG peaks decrease with citric acid dose ratio in region I, demonstrating that citric acid can enhance the degradation of susceptible organics in hydrothermal carbonization of sewage sludge. And the region III can be further subdivided into two intervals. In the range of 400–500 °C, the DTG peak intensity of AHC0.1 is the highest while the others are almost the same, but in the range of 500–600 °C, the DTG peak intensity of AHC0.5 is higher than those of the others. The interpretation can be that AHC0.1 contains more “fragile” aromatic compounds than SSHC and AHC0.5, which is the precursor to the “recalcitrant” aromatic compounds that exist abundantly in AHC0.5, proving that enhanced aromatization are obtained with increasing citric acid, which is in accordance with the elemental analysis results and proximate analysis results(seen in Table 1).

On the whole, the addition of citric acid improved the thermal and chemical stability of the produced hydrochars. The hydrochars are supposed to be relatively stable when temperature is under 200 °C, which makes them suitable for adsorption application under normal circumstances.

### 3.2. Adsorption Study

#### 3.2.1. Effect of Dose Ratio of Citric Acid

As illustrated in Figure 7, as the citric acid dose ratio increases from 0 to 0.1, the equilibrium adsorbed amount of Pb(II) on the SS-derived hydrochars also increase from 32.47 to 49.86 mg·g^−1^. Subsequently, the adsorbed amount decreases with the citric acid dose ratio (when it is larger than 0.1). The abrupt fall at the citric acid dose ratio of 0.5 might be caused by the reduction of adsorptive sites due to the decreased specific surface area (seen in Table 4), O-containing/N-containing groups (seen in Table 2 and Table 3) and inorganics (seen in Table 1). AHC0.1 prevails the others in the adsorption of Pb(II), which is primarily due to its higher contents of O-containing functional groups and higher value of specific surface area.

#### 3.2.2. Effect of Initial pH

The effect of initial pH on the adsorption of Pb(II) by AHC0.1 and SSHC was studied and illustrated in Figure 8a. The adsorbed amount of Pb(II) onto SSHC gradually increased as the increasing initial pH, and reached the maximum at a pH of 8. The equilibrium uptake of Pb(II) by AHC0.1 increases rapidly as the initial pH increases from 1 to 4, but remains relatively stable within the pH range of 4–8 with a range value of only 0.31 mg·g^−1^. Most importantly, when the pH is between 3 and 6, AHC0.1 outperforms SSHC in terms of Pb(II) adsorption.

It has been reported that the pH of the solution can affect the surface charge of adsorbent, the degree of ionization and the species of adsorbate ions, which have an important impact on the adsorption process [47]. SSHC and AHC0.1 have pH_pzc_ values of 6.07 and 5.34, respectively (seen in Table 2). The Pb(II) species at each initial pH calculated by Visual MINTEQ is shown in Figure 6b. When the pH of the solution is lower than the pH_pzc_ of the adsorbent (AHC0.1 and SSHC) (pH ≤ 5), the surface of the adsorbent becomes protonated, and the main species of Pb(II) in the solution are Pb^2+^ and Pb(NO_3_)^+^,there is electrostatic repulsion between them hindering the adsorption. As the initial pH of the solution increases gradually, the repulsion between the adsorbent and Pb(II) weakens, allowing for the enhanced adsorption. When the pH of the solution exceeds the pH_pzc_ of the adsorbent, the adsorbent’s surface de-protonates, and the predominant Pb species are Pb(OH)_2_, Pb^2+^, and Pb(OH)^+^, then the electrostatic attraction promotes the adsorption.

#### 3.2.3. Adsorption Kinetics

Pseudo-first-order model and pseudo-second order model were applied for the model-fitting of adsorption kinetics of Pb(II) by SSHC and AHC0.1 to the experimental data with the results depicted in Figure 9 and model parameters summarized in Table 5. In spite of that, both models fit the kinetics well with R^2^ values higher than 0.98, but the pseudo-first-order model provides the best fit for both SSHC (R^2^ = 0.998) and AHC0.1 (R^2^ = 0.999), implying that the adsorption of Pb(II) by SSHC and AHC0.1 could be chemical processes [48]. AHC0.1 prevails SSHC in terms of adsorption rate of Pb(II) at the very early stage of the adsorption for its higher value of K_2_, denoting the adsorption sites on SSHC are more available to Pb(II). Furthermore, the q_e_ of AHC0.1 (48.48 mg·g^−1^) is higher than that of SSHC (33.03 mg·g^−1^), which indicates that there are more adsorption sites for Pb(II) on AHC0.1. The results coincide with the larger specific area and richer O-containing functional groups of AHC0.1.

#### 3.2.4. Adsorption Isotherms

The adsorption isotherms of Pb(II) by AHC0.1 and SSHC were simulated using Langmuir and Freundlich models. The fitting results are exhibited in Figure 8 and the model parameters are listed in Table 6. Langmuir adsorption isotherm was developed based on the assumption that adsorption occur on the homogeneous surface without interaction between the adsorbate ions. Freundlich adsorption isotherm was used to describe the multi-layer adsorption on heterogeneous surface.

According to the fitting results presented in Figure 10 and Table 6, the Langmuir model fits the experimental data significantly better than Freundlich model. However, the BET model cannot converge to all the adsorption data, so the adsorption of Pb(II) onto AHC0.1 and SSHC was supposed to be monolayer. Hence, the adsorption is less likely to be governed by physisorption [34]. Considering the result of the mechanism study mentioned below, the adsorption process is prone to be governed by chemisorption, but physisorption also exists.

It is evident that AHC0.1′s affinity to Pb(II) at 298 K is stronger than that of SSHC for its higher value of K_L_ (Langmuir constant). Moreover, AHC0.1’s q_max_ (maximum adsorption capacity) of Pb(II) (60.88 mg·g^−1^, at 298 K) is also about 30% higher than that of SSHC (44.34 mg·g^−1^, at 298 K). Besides, the adsorption of Pb(II) onto SSHC and AHC0.1 can be regarded as endothermic process because all the values of K_L_ and q_max_ increase with adsorption temperature.

Although the specific surface area of AHC0.1 is relatively low (59.95 m^2^·g^−1^), but its maximum adsorption capacity of Pb(II) (60.88 mg·g^−1^) is superior or at least comparable to the majority of sewage sludge derived hydrochars, sewage sludge biochars and some hydrochars derived from other biomass (listed in Table 7), which might benefit from the rich O-containing/N-containing functional groups and inorganics (silicates and phosphates) in AHC0.1.

#### 3.2.5. Adsorption Thermodynamics

As depicted in Figure 11, ln(K) is linearly well-related with 1/T. The thermodynamic parameters are summarized in Table 8. For the adsorption of Pb(II) by AHC0.1 and SSHC, there exists: Δ*G* < 0, Δ*H >* 0, Δ*S >* 0, and |*T*Δ*S*| > Δ*H*, indicating that their adsorption of Pb(II) are spontaneous endothermic processes driven by entropy change. However, the adsorption enthalpy increase of SSHC is much higher than that of AHC0.1, which can be attributed to the lower absorbed heat for Pb(II) diffusion and the higher adsorption heat release in the adsorption of Pb(II) by AHC0.1 compared with SSHC due to its higher surface area and richer O-containing functional groups.

### 3.3. Adsorption Mechanism

As demonstrated in Figure 12, IER is in the range of 77.89% to 97.96% when the initial pH is between 2 and 6, indicating that the cation release (phenomenological ion exchange) contributes the most to the removal of Pb(II) by AHC0.1, and δ_Ca_I_Ca_ accounts for the largest proportion in Δ_qoc_ (Σδ_i_I_i_), and δ_Mg_I_Mg_ takes second place, denoting that the migration of Ca^2+^ and Mg^2+^ might play an important role. The released divalent and trivalent metal cations (Me) should mainly originate from complexes like −COO-Me or -Ar-O-Me and/or from phosphate/silicate precipitates.

When pH = 2, the adsorbed amount of Pb(II) is rather low, and the selectivity of exchanged cation decreased, manifested by the lowest proportion of δ_Ca_I_Ca_ in Δ_qoc_ within the whole pH range of 2–6, perhaps resulted from the dominated regeneration process caused by high concentration of H^+^ and the strong electrostatic repulsion [55]. And the value of IER is extremely close to 1 here, indicating that the phenomenological ion exchange domains, the fundamental mechanism might be co-precipitation/complexation with high bonding strength and cation-π interaction which is irrespective of pH change.

The value of Δ_qoc_ and the adsorbed amount of Pb(II) increase with initial pH, but the IER decreases, which can be interpreted by the weakening electrostatic repulsion and the preferentially enlarged exposure of aromatic C=C/C=N to Pb(II) which are potential adsorptive sites but non-essentially related to the phenomenological ion exchange.

Pb_5_SiO_7_ and Pb_2_P_2_O_7_ generated in the adsorption of Pb(II) by AHC0.1 were determined by XRD characterization (seen in Figure 5d), indicating that co-precipitation might play a significant role in the adsorption process. As illustrated in the DRIFT spectra of AHC0.1 and Pb@AHC0.1 (presented in Figure 1c,d), after the adsorption, the stretching vibration peak at 1057 cm^−1^ assigned to C−O−C migrated to 1061 cm^−1^ and its relative peak intensity significantly reduced; moreover, the out-of-plane C−H/N−H bending vibrational peak also significantly decreased. After AHC0.1 adsorbed Pb(II), the binding energy values of C=O (in carboxylates or aldehydes) and C−O/C−N increased by 0.69 and 0.12 eV, respectively, as depicted in Figure 2 and Table 3. Furthermore, the binding energy values of amine-N, protein-N, pyrrole-N, and −N=O also increased by 0.95, 1.00, 0.85, and 0.99 eV.

The DRIFT and XPS results indicate that O-containing groups(including C−O−C, C−O, and C=O), N-containing functional groups(including pyrrolic groups, −NH−, −NH_2_, and N=O) and aromatic groups play important part in the adsorption of Pb(II) by AHC0.1. Specifically, as depicted in Figure 13, C−O and C=O can adsorb Pb^2+^ by complexation or coordination; C−O−C, −NH−, −NH_2_, and N=O can coordinate with Pb^2+^, and aromatic groups and pyrrolic groups can adsorb Pb^2+^ through cation-π interaction. In summary, the adsorption of Pb(II) by AHC0.1 is widely influenced by complexation, precipitation, cation-π interaction, and electrostatic interaction.

## 4. Conclusions

A cost-effective adsorbent for Pb(II) was prepared by hydrothermal carbonization using sewage sludge and inexpensive citric acid as feedstock. The introduction of citric acid into the preparation of sewage sludge-derived hydrochar can effectively develop its pore structure, increase its surface functional groups, and improve its thermal and chemical stability. Demonstrated by the result of adsorption experiments, AHC0.1 has great adsorption ability for its fast adsorption kinetics and maximum adsorption capacity of 60.88 mg·g^−1^, which is competitive to other adsorbents. AHC0.1 adsorbs and removes Pb(II) mainly through its phenomenological ion exchange with other metal cations especially Ca^2+^ and Mg^2+^. Further, the adsorption can be mainly attributed to co-precipitation, complexation, cation-π interaction, and electrostatic interaction. Due to the low-cost feedstock, the facile production, and the significant potential of the produced hydrochar as adsorbents for heavy metal removal, the joint hydrothermal carbonization of citric acid and sewage sludge can achieve the value-added utilization of sewage sludge as adsorbents.

## Figures and Tables

**Figure 1 polymers-14-00968-f001:**
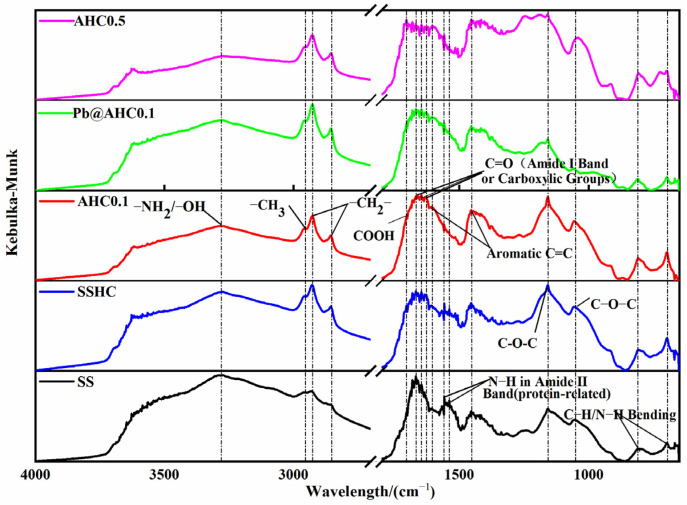
DRIFT spectra of SS, SS-derived hydrochars, and Pb@AHC0.1.

**Figure 2 polymers-14-00968-f002:**
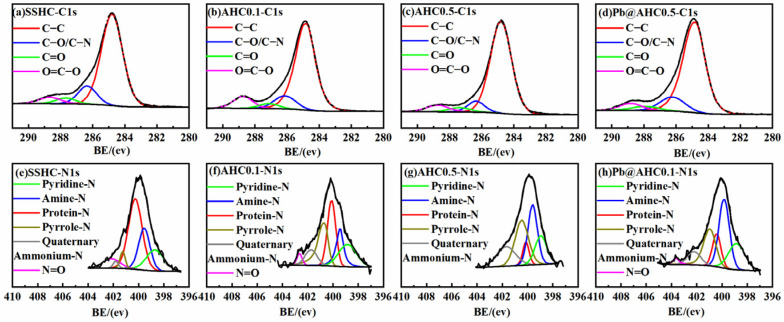
C1s and N1s XPS spectra of SS-derived hydrochars and Pb@AHC0.1.

**Figure 3 polymers-14-00968-f003:**
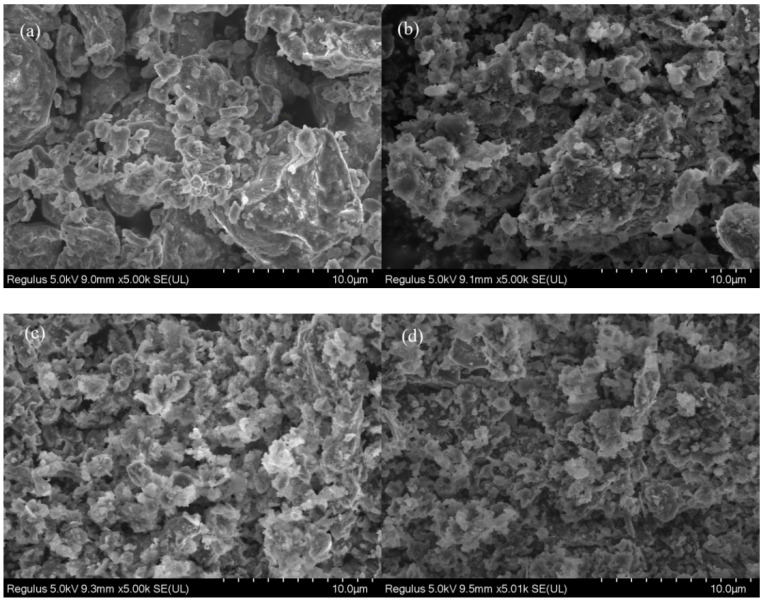
SEM image of dry sewage sludge (**a**), SSHC (**b**), AHC0.1 (**c**), and AHC0.5 (**d**).

**Figure 4 polymers-14-00968-f004:**
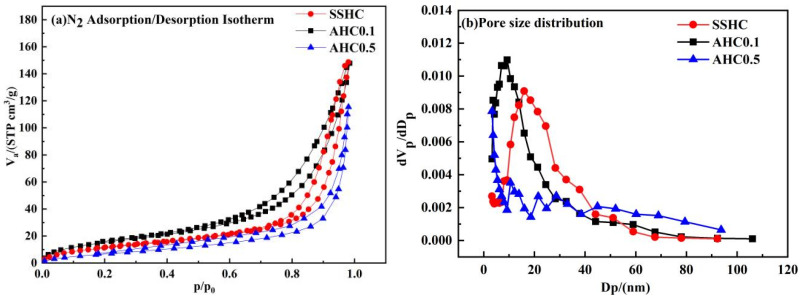
N_2_ adsorption/desorption isotherm plots (**a**) and pore size distribution curves (**b**).

**Figure 5 polymers-14-00968-f005:**
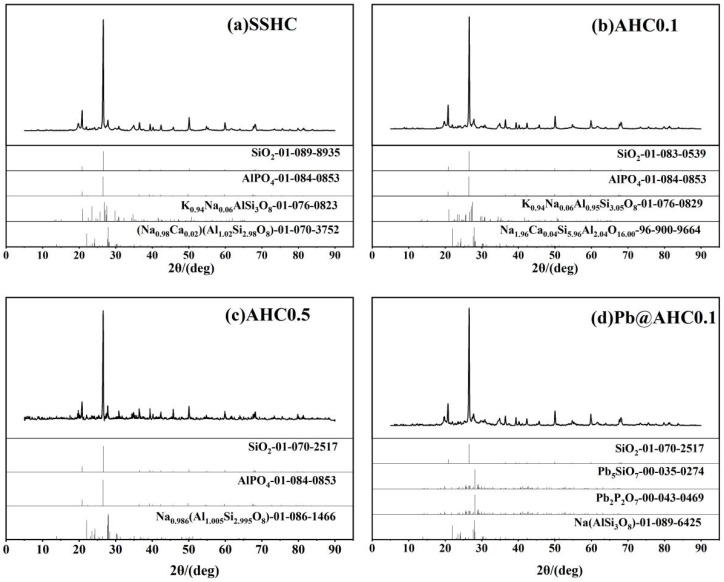
XRD patterns of SS-derived hydrochars and Pb@AHC0.1.

**Figure 6 polymers-14-00968-f006:**
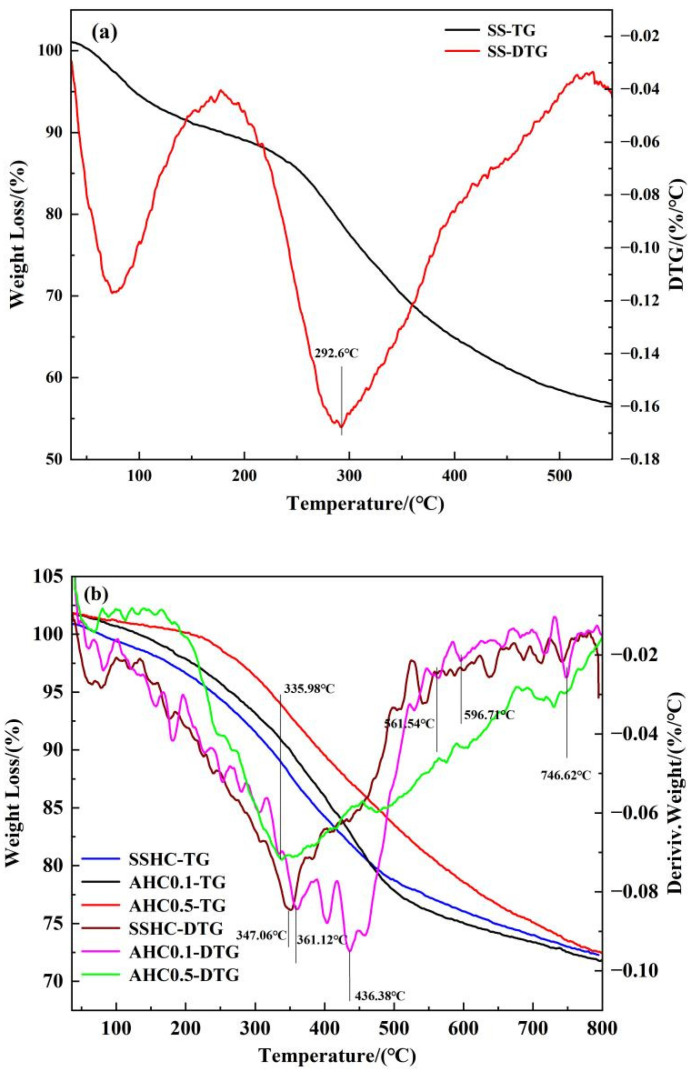
TG/DTG curves of SS (**a**) SSHC, AHC0.1, and AHC0.5 (**b**).

**Figure 7 polymers-14-00968-f007:**
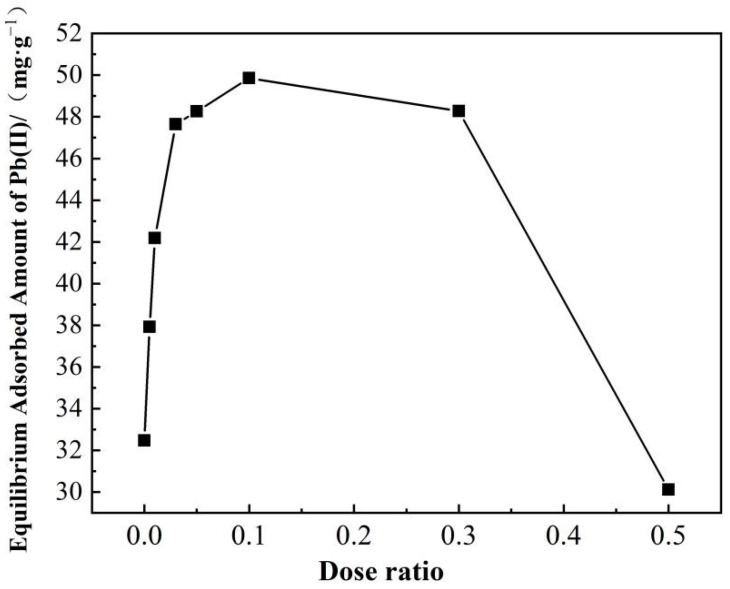
Effects of dose ratio of citric acid on the adsorption of Pb(II) onto AHCs.

**Figure 8 polymers-14-00968-f008:**
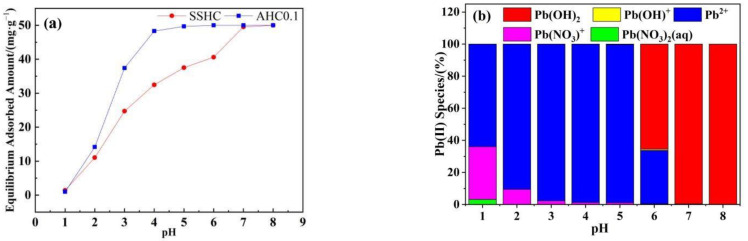
Effects of initial pH on Pb(II) adsorption (**a**) and Pb species under different pH (**b**).

**Figure 9 polymers-14-00968-f009:**
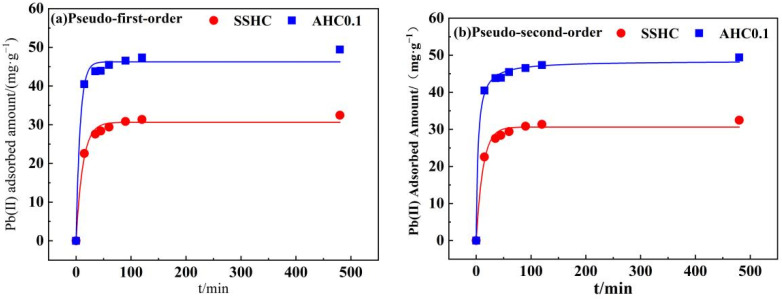
Non-linear fitting of Pseudo-first-order model (**a**) and Pseudo-second-order model (**b**).

**Figure 10 polymers-14-00968-f010:**
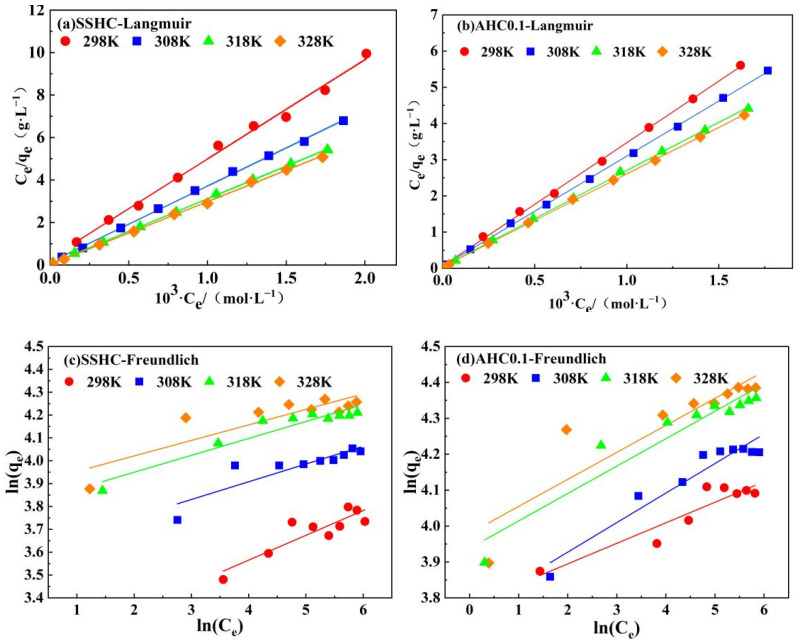
Linear fitting of experimental data using Langmuir model and Freundlich model.

**Figure 11 polymers-14-00968-f011:**
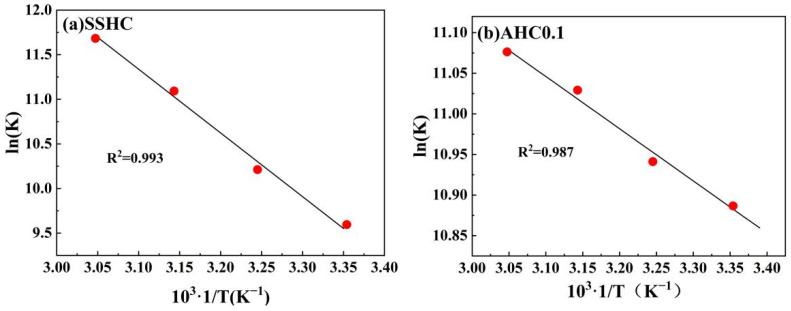
Linear regressions of van’t Hoff plot.

**Figure 12 polymers-14-00968-f012:**
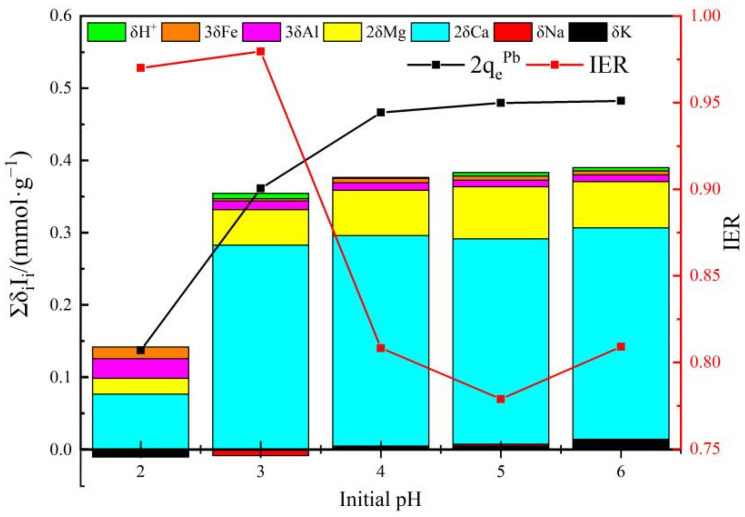
Phenomenological ion exchange in the adsorption of Pb(II) by AHC0.1.

**Figure 13 polymers-14-00968-f013:**
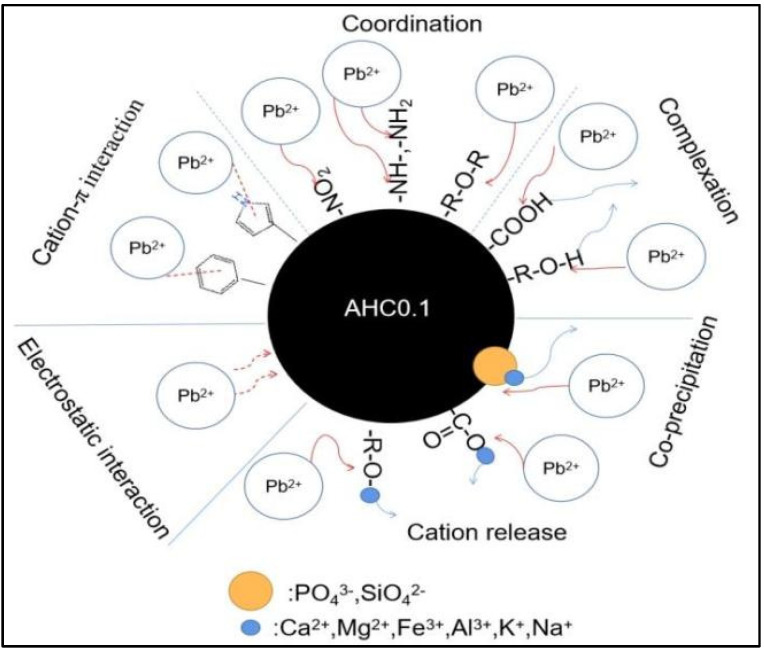
Schematic of Pb(II) adsorption mechanism by AHC0.1.

**Table 1 polymers-14-00968-t001:** Ultimate analysis and proximate analysis of SS and SS-derived hydrochars.

	SS	SSHC	AHC0.1	AHC0.5
C (%)	26.53	19.45	18.64	21.65
H (%)	6.47	3.23	2.99	2.78
O (%)	17.42	10.13	11.88	9.26
N (%)	4.72	1.58	1.39	1.22
S (%)	0.72	0.48	0.42	0.49
H/C	2.93	1.99	1.92	1.54
O/C	0.49	0.39	0.48	0.32
Volatile Content (%)	50.19	29.11	30.62	28.71
Fixed Carbon Content (%)	5.66	5.82	4.62	6.69
Ash Content (%)	44.15	65.07	64.76	64.60

**Table 2 polymers-14-00968-t002:** Content of oxygen-containing groups and pH_pzc_ of SS-derived hydrochars.

Hydrochars	Phenolic Groups ueq·g^−1^	Lactone Groups ueq·g^−1^	Carboxyl Groups ueq·g^−1^	pH_pzc_
SSHC	350	316	83	6.07
AHC0.1	400	700	450	5.34
AHC0.5	167	373	110	6.05

**Table 3 polymers-14-00968-t003:** Deconvolution results of C1s and N1s spectra.

	SSHC		AHC0.1		AHC0.5		Pb@AHC0.1	
Species	BE ev	Contents at.%	BE ev	Contents at.%	BE ev	Contents at%	BE ev	Contents at.%
C−C	284.80	76.55	284.80	74.91	284.80	82.03	284.80	76.88
C−O/C−N	286.34	13.33	286.11	11.53	286.34	7.84	286.23	13.00
C=O	287.67	4.55	287.12	4.00	287.38	3.91	287.91	4.25
O=C−O	288.67	5.57	288.75	9.56	288.74	6.22	288.81	5.87
Pyridine-N	398.70	17.55	398.85	19.83	398.94	16.75	398.83	12.98
Amine-N	399.55	22.63	399.42	16.14	399.60	28.03	399.80	34.18
Protein-N	400.26	46.03	400.11	27.32	400.13	7.26	400.42	22.36
Pyrrole-N	401.26	4.21	400.69	24.22	401.46	29.88	400.96	19.96
Quartenary Ammonium-N	401.81	1.77	401.71	9.35	401.69	18.08	402.21	8.22
N=O	402.14	7.82	402.65	3.13	-	-	403.64	2.31

**Table 4 polymers-14-00968-t004:** Pore characteristics of SS-derived hydrochars.

SS-Derived Hydrochars	Specific Surface Area m^2^·g^−1^	Total Pore Volume cm^3^·g^−1^	Average Pore Size nm	Peak Pore Size nm
HC	45.53	0.23	20.21	12.12
AHC0.1	59.95	0.23	15.25	9.23
AHC0.5	30.10	0.18	23.74	3.29

**Table 5 polymers-14-00968-t005:** Adsorption kinetics parameters of Pb(II) onto SSHC and AHC0.1.

SS-Derived Hydrochars	Pseudo-First-Order Model			Pseudo-Second-Order Model		
	K_1_	q_e_/(mg·g^−1^)	R^2^	K_2_	q_e_/(mg·g^−1^)	R^2^
SSHC	0.08	30.63	0.988	0.004	33.03	0.999
AHC0.1	0.13	46.24	0.989	0.006	48.48	0.998

**Table 6 polymers-14-00968-t006:** Adsorption isotherms parameters of Pb(II) onto SSHC and AHC0.1.

SS-Derived Hydrochars	Temperature K	Langmuir Model			Freundlich Model		
		q_max_ mg·g^−1^	K_L_ L·mg^−1^	R^2^	n	K_F_	R^2^
SSHC	298	44.34	0.07	0.981	9.21	22.90	0.762
	308	57.79	0.13	0.999	12.80	12.89	0.753
	318	67.46	0.32	0.999	13.52	13.52	0.892
	328	70.05	0.57	0.999	14.73	14.73	0.723
AHC0.1	298	60.88	0.26	0.999	17.48	43.82	0.872
	308	68.09	0.27	0.999	12.16	43.07	0.924
	318	77.99	0.30	0.999	13.10	51.30	0.895
	328	80.73	0.31	0.999	13.30	53.49	0.876

**Table 7 polymers-14-00968-t007:** Comparison of maximum adsorption capacity of Pb(II) onto various adsorbents.

Adsorbents	Q_max_ (mg·g^−1^)	S_BET-N__2_ (m^2^·g^−1^)	Reference
H_2_O_2_ modified peanut hull hydrochar	22.82	1.4	[49]
SiO_2_/polysaccharide-hydrochar composite	52	21.76	[50]
LDH/sewage sludge-hydrochar composite	62.44	-	[51]
KOH activated sewage sludge biochar	57.48	907.95	[52]
Sewage sludge-based biochar	18.20	23.70	[53]
Sewage sludge-based biochar	51.20	-	[54]
SSHC	44.34	45.53	This study
AHC0.1	60.88	59.95	This study

**Table 8 polymers-14-00968-t008:** Parameters of adsorption thermodynamics.

Hydrochars	T (K)	ΔH (KJ·mol^−1^)	ΔS (J·mol^−1^·L^−1^)	ΔG (KJ·mol^−1^)
SSHC	298	57.90	274.15	−23.68
	308			−26.42
	318			−29.16
	328			−31.90
AHC0.1	298	5.35	108.41	−26.99
	308			−28.03
	318			−29.17
	328			−30.22

## Data Availability

Not applicable.

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
