# Peer review of "Preparation of Citric Acid-Sewage Sludge Hydrochar and Its Adsorption Performance for Pb(II) in Aqueous Solution"

_polymers, 2022, doi:10.3390/polym14050968_

Round 1

Reviewer 1 Report

Manuscript ID: polymers-1597125

I have reviewed the manuscript entitled “Preparation of citric acid-sewage sludge hyrochar and its adsorption performance for Pb(II) in aqueous solution”, Manuscript needs serious revision. My specific comments as follow:

Typo error in title, check the spelling of “Hydrochar”.

Abstract is ok.

In introduction authors should cite the following study on sewage sludge: 1) Roof solar drying processes for sewage sludge within sandwhich-like chamber bed. Renewable Energy (2019) 136: 1071-1081; 2) A new drying kinetic model for sewage sludge drying in presence of CaO and NaClO. Applied Thermal Engineering 2016, 106: 141-152.

The objectives of the research should be clearly mentioned in the last paragraph of the introduction.

In the methods section, the volume of the adsorbate should be clearly mentioned in section 2.4.1 & 2.4.2. “Dm” should be explained here.

Use correct reference for Eq. (13), Comments on “Reasonable calculation of the thermodynamic parameters from adsorption equilibrium constant, Journal of Molecular Liquids 322 (2021) 114980.”. J Mol Liq, 334, 116542.

Section 3.1.4, authors should give the Nitrogen adsorption-desorption isotherm plot for surface area calculation.

Sec. 3.2.1 Authors should justify, why Pb adsorption suddenly falls at citric acid dose ratio of 0.5?

Authors use the term “large surface area” which is not suitable for an adsorbent that has a surface area of 30.10 to 59.95 m2/g.

The authors should explain, why the adsorption of Pb(II) was 60.99 mg/g, even though the adsorbent surface area was <60 m2/g.

Sec. 3.3 Mechanism of the adsorption should be illustrated with some graphical or chemical reaction.

The conclusion should reflect the objective of this research.

Author Response

Responses to the reviewers’ comments on “Preparation of citric acid-sewage sludge hydrochar and its adsorption performance for Pb(Ⅱ) in aqueous solution” (Manuscript ID: Polymers-1597125)

Dear reviewer:

We really appreciated you for your time and valuable suggestions on our manuscript. We have taken very careful considerations on your comments to improve our manuscript as possible as we can. In the revised manuscript, all the corrections and changes have been labeled in yellow. The detailed responses are as follows:

  1. Typo error in title, check the spelling of “Hydrochar”;

Answer: The typo error in title has been corrected. Please check line 1-2 in the revised manuscript.

  1. In introduction, authors should cite the following study on sewage sludge:
  • Roof solar drying processes for sewage sludge within sandwhich-like chamber bed. Renewable Energy (2019) 136: 1071-1081;
  • A new drying kinetic model for sewage sludge drying in presence of CaO and NaClO. Applied Thermal Engineering 2016, 106: 141-152.

Answer: According to your suggestions, the related publications has been cited.

Traditional thermal treatments such as incineration, pyrolysis, and gasification can recover the heat from sewage sludge, but the feedstock needs pre-drying. Nowadays, many novel drying methods, like roof solar drying methods[6] and chemical-assisted drying methods[7], have been developed, which can solve the problem of high energy consumption of the conventional thermal drying methods. But the traditional thermal treatments are still not without drawbacks such as high complexity of the system, high construction cost, and the release of secondary environmental pollutants (NOx, SOx and dioxins etc)[8].

[6]    Wang, P., D. Mohammed, P. Zhou, Z. Lou, P. Qian, and Q. Zhou," Roof solar drying processes for sewage sludge within sandwich-like chamber bed", RENEW ENERG Vol. 136, 2019, pp. 1071-1081.

[7]    Danish, M., H. Jing, Z. Pin, L. Ziyang, and Q. Pansheng," A new drying kinetic model for sewage sludge drying in presence of CaO and NaClO", APPL THERM ENG Vol. 106, 2016, pp. 141-152.

[8]    Liu, T., Y. Li, N. Peng, Q. Lang, Y. Xia, C. Gai, Q. Zheng, and Z. Liu," Heteroatoms doped porous carbon derived from hydrothermally treated sewage sludge: Structural characterization and environmental application", J ENVIRON MANAGE Vol. 197, 2017, pp. 151-158.

Please check Line 36-42 in the revised manuscript.

  1. The objectives of the research should be clearly mentioned in the last paragraph of the introduction.

Answer: The objectives of the research has been clearly clarified in the revised manuscript as follow:

Thus, in this study, the hydrochars were prepared through the co-hydrothermal carbonization of sewage sludge and citric acid and applied to eliminate Pb(Ⅱ) from aqueous solution. The main objective of this study was to provide some insight into the effect of citric acid addition in the hydrothermal carbonization of sewage sludge on the adsorbents-related physicochemical properties and the heavy metal adsorption performance of the produced hydrochars. For this purpose, the specific tasks of this research was to (1)characterize the produced hydrochars by elemental analysis, proximate analysis, diffuse reflectance fourier transform infrared spectroscopy(DRIFT), X-ray photoelectron spectrascopy(XPS), pHpzc titration, Boehm titration, X-ray diffraction spectroscopy(XRD), N2 adsorption/desorption isdotherms, scanning electron microscopy(SEM) and thermogravimetric analysis(TGA); (2) investigate and compare the adsorption of Pb(Ⅱ) onto the hydrochars by batch experiments; (3) acquire the further insight into the potential adsorption mechanism.

Please check the yellow part in Line 96-106 in the revised manuscript..

  1. In the methods section, the volume of the adsorbate should be clearly mentioned in section 2.4.1 & 2.4.2. “Dm” should be explained here.

Answer: The volume of the adsorbates and dosage of hydrochars has been given as: The volume of adsorbates and the dosage of hydrochars were setted as 50 mL and 0.1 g respectively for all the experiments. Please check the yellow parts in Line 157-158 in the revised manuscript.

Dm has been explained as citric acid dose ratio in the AHC preparation. Please check the yellow parts in Line149 in the revised manuscript.

  1. Use correct reference for Eq. (13), Comments on “Reasonable calculation of the thermodynamic parameters from adsorption equilibrium constant, Journal of Molecular Liquids 322 (2021) 114980.”. J Mol Liq, 334, 116542.

Answer: The reference has been corrected as: Since it was proved that the adsorption of SSHC and AHC0.1 fitted Langmuir model well , then [36]:

[36]  Lima, E.C., F. Sher, M.R. Saeb, M. Abatal, and M.K. Seliem," Comments on “Reasonable calculation of the thermodynamic parameters from adsorption equilibrium constant, Journal of Molecular Liquids 322 (2021) 114980.”", J MOL LIQ Vol. 334, 2021, pp. 116542.

Please check in the yellow part in Line 211-212 in the revised manuscript.

  1. Section 3.1.4, authors should give the Nitrogen adsorption-desorption isotherm plot for surface area calculation.

Answer: The nitrogen adsorption-desorption isotherm plots have been given in the revised manuscript as Fig.4(a):

Fig.4(a) N2 adsorption/desorption isotherm plots

Please check the yellow parts in Line 324-326 in the revised manuscript.

And the corresponding interpretation has been given as: The N2 adsorption/desorption isotherm plots can be classified as type Ⅳ isotherm with H3 hysteresis loop[46], which indicates slit mesopores.

[46]  Sing, K.S.W., and R.T. Williams," Physisorption Hysteresis Loops and the Characterization of Nanoporous Materials", ADSORPT SCI TECHNOL Vol. 22, No. 10, 2004, pp. 773-782.

Please Check the yellow parts in Line 315-316,in Page10.

  1. 3.2.1 Authors should justify, why Pb adsorption suddenly falls at citric acid dose ratio of 0.5?

Answer: The justification has been given as: Subsequently, the adsorbed amount decreases with the citric acid dose ratio(when it’s larger than 0.1). The abrupt fall at the citric acid dose ratio of 0.5 might be caused by the reduction of adsorptive sites due to the decreased specific surface area(seen in Table4), O-containing/N-containing groups(seen in Table2 and Table3) and inorganics(seen in Table1).

Please check the yellow parts in Line 370-374 in the revised manuscript.

  1. Authors use the term “large surface area” which is not suitable for an adsorbent that has a surface area of 30.10 to 59.95 m2/g.

Answer: Indeed, the description was inaapropriate and the term has been deleted. Please check the revised manuscript.

  1. The authors should explain, why the adsorption of Pb(II) was 60.99 mg/g, even though the adsorbent surface area was <60 m2/g.

Answer: The explanation has been given as: Although the specific surface area of AHC0.1 is relatively low(59.95 m2·g-1), but its maximum adsorption capacity of Pb(Ⅱ) (60.88 mg·g-1) is superior or at least comparable to that of the majority of sewage sludge drived hydrochars, sewage sludge biochars and some hydrochars derived from other biomass(listed in Table7), which might benefit from the rich O-containing/N-containing functional groups and inorganics(silicates and phosphates) in AHC0.1.

Table 7 Comparison of maximum adsorption capacity of Pb(Ⅱ) onto various adsorbents

Adsorbents

Qmax  (mg·g-1)

SBET-N2 (m2·g-1)

Reference

H2O2 modified peanut hull hydrochar

22.82

1.4

[50]

SiO2/polysaccharide-hydrochar composite

52

21.76

[51]

LDH/sewage sludge-hydrochar composite

62.44

-

[52]

KOH activated sewage sludge biochar

57.48

907.95

[53]

Sewage sludge-based biochar

18.20

23.70

[54]

Sewage sludge-based biochar

51.20

-

[55]

SSHC

44.34

45.53

This study

AHC0.1

60.88

59.95

This study

[50]  Xue, Y., B. Gao, Y. Yao, M. Inyang, M. Zhang, A.R. Zimmerman, and K.S. Ro," Hydrogen peroxide modification enhances the ability of biochar (hydrochar) produced from hydrothermal carbonization of peanut hull to remove aqueous heavy metals: Batch and column tests", CHEM ENG J Vol. 200-202, 2012, pp. 673-680.

[51]  Li, Y., K. Li, M. Su, Y. Ren, Y. Li, J. Chen, and L. Li," Fabrication of carbon/SiO2 composites from the hydrothermal carbonization process of polysaccharide and their adsorption performance", CARBOHYD POLYM Vol. 153, 2016, pp. 320-328.

[52]  Luo, X., Z. Huang, J. Lin, X. Li, J. Qiu, J. Liu, and X. Mao," Hydrothermal carbonization of sewage sludge and in-situ preparation of hydrochar/MgAl-layered double hydroxides composites for adsorption of Pb(II)", J CLEAN PROD Vol. 258, 2020, pp. 120991.

[53]  Zhang, J., J. Shao, Q. Jin, Z. Li, X. Zhang, Y. Chen, S. Zhang, and H. Chen," Sludge-based biochar activation to enhance Pb(II) adsorption", FUEL Vol. 252, 2019, pp. 101-108.

[54]  Zhang, W., S. Mao, H. Chen, L. Huang, and R. Qiu," Pb(II) and Cr(VI) sorption by biochars pyrolyzed from the municipal wastewater sludge under different heating conditions", BIORESOURCE TECHNOL Vol. 147, 2013, pp. 545-552.

[55]  Ho, S., Y. Chen, Z. Yang, D. Nagarajan, J. Chang, and N. Ren," High-efficiency removal of lead from wastewater by biochar derived from anaerobic digestion sludge", BIORESOURCE TECHNOL Vol. 246, 2017, pp. 142-149.

Please check in yellow parts in Line 441-448 in the revised manuscript.

  1. Sec. 3.3 Mechanism of the adsorption should be illustrated with some graphical or chemical reaction.

Answer: The mechanism of the adsorption has been illustrated in Fig.13 as follow:

Fig.13 Schematic of Pb(Ⅱ) adsorption mechanism by AHC0.1

Please check the yellow parts in Line 483-485 in the revised manuscript.

  1. The conclusion should reflect the objective of this research.

Answer: The conclusion has been improved as follow:

A cost-effective adsorbent for Pb(Ⅱ) was prepared by hydrothermal carbonization using sewage sludge and inexpensive citric acid as feedstock. The introduction of citric acid into the preparation of sewage sludge-derived hydrochar can effectively develop its pore structure, increase its surface functional groups and improve its thermal and chemical stability. Demonstrated by the result of adsorption experiments, AHC0.1 has great adsorption ability for its fast kinetics and maximum adsorption capacity of 60.88 mg·g-1 which is competitive to other adsorbents. AHC0.1 adsorbs and removes Pb(Ⅱ) mainly through its phenomenological ion exchange with other metal cations especially Ca2+ and Mg2+. And the adsorption can be mainly attributed to co-precipitation, complexation, cation-π interaction and electrostatic interaction. Due to the low-cost feedstock, the facile production and the significant potential of the produced hydrochar as adsorbents for heavy metal removal, the joint hydrothermal carbonization of citric acid and sewage sludge can achieve the value-added utilization of sewage sludge.

Please check the yellow parts in Line 504-515 in the revised manuscript.

Reviewer 2 Report

Title : Preparation of Citric Acid-Sewage Sludge Hydrohcar and Its Adsorption Performance for Pb(Ⅱ) in Aqueous Solution

The work describes about utilization of sewage sludge and development of low-cost and high-efficient adsorbents, sewage sludge was hydrothermally carbonized with citric acid to prepare the hydrochar. The differences in physicochemical properties involving material compo-sition (ultimate analysis, XPS), surface functional groups (DRIFT, Boehm titration), mineral compo-sition (XRD), and pore characteristics (N2 adsorption/desorption) as well as Pb(II) adsorption per-formance between the obtained citric acid-sewage sludge hydrochars (AHC) and the hydrochar prepared solely from sewage sludge (SSHC) were investigated. When citric acid dose ratio (mass ratio of citric acid to dry sewage sludge) is 0.1, the obtained hydrohcar (AHC0.1) has the highest specific surface area (59.95 m2•g-1), the most abundant oxygen-containing functional groups, the lowest pHpzc (5.43), and the highest equilibrium adsorption capacity for Pb(II) (pH=4).The maximum adsorption capacity of AHC0.1 for Pb(II) is 60.88 mg•g-1 (298 K), which is approximately 1.3 times that of SSHC. The main mechanisms of Pb(II) adsorption by AHC0.1 are electrostatic attraction, precipitation, complexation, and cation-π interaction. It was demonstrated that by incorporating citric acid into the hydrothermal carbonization, resource utilization of sewage sludge can be accom-plished effectively. The work is interesting but needs major changes see below comments:

  1. Abstract: Its too verbose present some prominent only.
  2. Introduction: To highlight the importance of adsorption in introduction please lay stress on other methods such as ion exchange, photocatalysis etc then discus their importance as adsorption can follow Journal of Hazardous Materials 416 (2021), 125714.
  3. Please introduce some tables related to properties of their material.
  4. If possible, also discuss the thermal and chemical stability of material.
  5. If possible, please add some SEM or images of material to explore surface heterogeneity.
  6. Rediscus adsorption isotherms in detail clearly state which type of adsorption in nature physical or chemical see Gels 8 (2022), 23 org/10.3390/gels8010023.
  7. Please discus your ion exchange study in detail way in manuscript see Ionics 21 (2015), 1045-1055
  8. What functional groups are present on surface of material which are responsible for adsorption.
  9. In conclusion section state why, your material is better adsorbent.
  10. Compare your adsorbent results with other reported work.
  11. Improve the quality of figures specifically XRD.
  12. If possible, provide pictorial presentation showing mechanism about adsorption.

Author Response

Responses to the reviewers’ comments on “Preparation of citric acid-sewage sludge hydrochar and its adsorption performance for Pb(Ⅱ) in aqueous solution” (Manuscript ID: Polymers-1597125)

Dear reviewer:

We really appreciated you for your time and valuable suggestions on our manuscript. We have taken very careful considerations on your comments to improve our manuscript as possible as we can. In the revised manuscript, all the corrections and changes have been labeled in yellow. The detailed responses are as follows:

  1. Abstract: Its too verbose, present some prominent only.

Answer: The abstract has been improved as follow: In order to seek the value-added utilization method of sewage sludge and develop low-cost and high-efficient adsorbents, the hydrochar was prepared by the co-hydrothermal carbonization of sewage sludge and citric acid and characterized. The differences in Pb(II) adsorption performance between the citric acid-sewage sludge hydrochars (AHC) and the hydrochar prepared solely from sewage sludge (SSHC) were also investigated. When citric acid dose ratio (mass ratio of citric acid to dry sewage sludge) is 0.1, the obtained hydrohcar (AHC0.1) has the highest specific surface area (59.95 m2•g-1), the most abundant oxygen-containing functional groups, the lowest pHpzc (5.43), and the highest equilibrium adsorption capacity for Pb(II).The maximum adsorption capacity of AHC0.1 for Pb(II) is 60.88 mg•g-1 (298 K), which is approximately 1.3 times that of SSHC. The potential mechanisms can be electrostatic attraction, co-precipitation, complexation, and cation-π interaction. It was demonstrated that by incorporating citric acid into the hydrothermal carbonization, resource utilization of sewage sludge can be accomplished effectively.

Please check the yellow part in Line 12-22 in the revised manuscript.

  1. Introduction: To highlight the importance of adsorption in introduction please lay stress on other methods such as ion exchange, photocatalysis etc then discus their importance as adsorption can follow Journal of Hazardous Materials 416 (2021), 125714. Please introduce some tables related to properties of their material.

Answer: Water pollution caused by heavy metals(especially Pb, Cr, Cd and so on) has become a prominent challenge. Numerous treatment methods have been investigated and applied for heavy metal removal from wastewater, including adsorption, membrane methods (ultra-,nano-,micro-filtration, reverse osmosis, and eletrodialysis), chemical method (precipitation, coagulation&flocculation and flotation), ion exchange, electrochemical methods, photocatalytic methods and coupled methods like adsorptional photocatalysis[15-18]. Compared with the others, adsorption has the advantages of easy operation, low cost and high removal efficiency. Developing high efficient and cost-effective adsorbents has attracted much attention and research intreast, especially carbonaceous adsorbents prepared through thermo-chemical transformation of biowastes.

[15] Sharma, G., A. Kumar, M. Naushad, B. Thakur, D.N. Vo, B. Gao, A.A. Al-Kahtani, and F.J. Stadler," Adsorptional-photocatalytic removal of fast sulphon black dye by using chitin-cl-poly(itaconic acid-co-acrylamide)/zirconium tungstate nanocomposite hydrogel", J HAZARD MATER Vol. 416, 2021, pp. 125714.

[16] Sharma, G., D. Pathania, M. Naushad, and N.C. Kothiyal," Fabrication, characterization and antimicrobial activity of polyaniline Th(IV) tungstomolybdophosphate nanocomposite material: Efficient removal of toxic metal ions from water", CHEM ENG J Vol. 251, 2014, pp. 413-421.

[17] Qasem, N.A.A., R.H. Mohammed, and D.U. Lawal," Removal of heavy metal ions from wastewater: a comprehensive and critical review", npj Clean Water Vol. 4, No. 1, 2021, pp. 36.

[18] Fu, F., and Q. Wang," Removal of heavy metal ions from wastewaters: A review", J ENVIRON MANAGE Vol. 92, No. 3, 2011, pp. 407-418.

Please check the yellow part in Line 58-66 in the revised manuscript.

  1. If possible, also discuss the thermal and chemical stability of material.

Answer: The thermal and chemical stability was discussed using TG/DTG in the section 3.1.7:

Fig.6 TG/DTG curves of SS(a) SSHC, AHC0.1 and AHC0.5(b)

TG under N2 atmosphere was conducted and DTG curves were also considered for themal properties of the hydrochars.The DTG curves can be divided as four weight loss regions. The first interval (region Ⅰ) ranges from 0℃ to 200℃ can be typically ascribed to the release of free water, bound water and low-boiling-point organic compounds. Mass loss in the temperature interval of 200℃-400℃(region Ⅱ) can be attributed to the degradation of susceptile matters, including organic acids, alkyl moiety and carbohydrates[47]. Region Ⅲ spans a temperature range of 400℃-600℃, of which weight loss resulted from the decomposition of aromatic compounds. And the transformation of inorganics occured in the temperature interval of 600-800(region Ⅳ ). Apparently, the peak temperature in the DTG curve of SS is much lower than that of SSHC, with its DTG peak intensity much higher than hydrochar’s, indicating that hydrothermal carbonization can improve thermal stability. As seen in Fig.6(b), the intensity of DTG peaks decrease with citric acid dose ratio in region Ⅰ, demonstrating that citric acid can enhance the degradation of susceptile organics in hydrothermal carbonization of sewage sludge. And the region Ⅲ can be further subdivided into 2 intervals. In 400℃-500℃, the DTG peak intensity of AHC0.1 is the highest with the others’ are almost the same, but in the range of 500℃-600℃, the DTG peak intensity of AHC0.5 is higher than those of the others. The interpretation can be that AHC0.1 contains more “fragile” aromatic compounds than SSHC and AHC0.5, which is the precursor to the “recalcitrant” aromatic compounds that exist abundantly in AHC0.5, proving that enhanced aromatization are obtained with increasing citric acid, which is in accordance with the proximate analysis results.

On the whole, addition of citric acid improved the thermal and chemical stability of the produced hydrochars.The hydrochars are supposed to be ralatively stable when temperature is under 200℃, which makes them suitable for adsorption application under normal circumstances.

Please check the yellow part in Line 341-363 in the revised manuscript.

  1. If possible, please add some SEM or images of material to explore surface heterogeneity.

Answer: The SEM images has been given in Fig.3 in the revised manuscript.

Fig.3 SEM image of dry sewage sludge(a), SSHC(b), AHC0.1(c) and AHC0.5(d)

SEM was applied to study the morphology evolution from sewage sludge to hydrochars. As depicted in the Fig.3(a), dry sewage sludge can be regarded as clustered aggregates with smooth surface and few pores. Compared with dry sewage sludge, SSHC exhibits much rougher surface with more pores, which might be caused by the release of some volatile contents. This also indicates that hydrothermal carbonization can develop pore structure. In comparison with SSHC, AHC0.1 and AHC0.5 display more fragments, flakes and sphere-like microparticles adhered. It can be deduced that addition of citric acid can promote the degradation of macro-organics and increase the carbonization degree. But the effect of citric acid addition on the pore characteristics need quantative analysis through N2 adsorption/desorption isotherms.

Please check in Line 298-310 in the revised manuscript.

  1. Rediscus adsorption isotherms in detail clearly state which type of adsorption in nature physical or chemical see Gels 8 (2022), 23 org/10.3390/gels8010023.

Answer: The discussion has been given as: BET isotherm model, as a multilayer adsorption isotherm model, was also considered to simulate the data of Pb() adsorption by AHC0.1 and SSHC[34, 35]:

 (9)

Where Cs stands for monolayer saturation concentration, mg·L-1, CBET denotes BET constants, L·mg-1,qs represents isotherm saturation capacity, mg·g-1.

please

According to the fitting results presented in Fig.10 and Table6, the Langmuir model fits the experimental data significantly better than Freundlich model. But BET model can’t converge to all the adsorption data, so the adsorption of Pb() onto AHC0.1 and SSHC was supposed to be monolayer. Hence,the adsoroption is less likely to be governed by physisorption[34]. Considering the result of mechanism study mentioned below, the adsorption process is prone to be governed by chemisorption,but physisorption also exists .

[34]  Sharma, G., A. Kumar, A.A. Ghfar, A. García-Peñas, M. Naushad, and F.J. Stadler, "Fabrication and Characterization of Xanthan Gum-cl-poly(acrylamide-co-alginic acid) Hydrogel for Adsorption of Cadmium Ions from Aqueous Medium", Gels, 2022.

[35]  Ebadi, A., J.S. Soltan Mohammadzadeh, and A. Khudiev," What is the correct form of BET isotherm for modeling liquid phase adsorption?", ADSORPTION Vol. 15, No. 1, 2009, pp. 65-73.

Please check in Line 204-208 and Line432-436 in the revised manuscript.

  1. Please discus your ion exchange study in detail way in manuscript see Ionics 21 (2015), 1045-1055

Answer: The discussion has been given as: When pH=2 , the adsorbed amount of Pb(Ⅱ) is rather low, and the selectivity of exchanged cation  decreased, manefested by the lowest proportion of δCaICa in Δqoc within the whole pH range of 2-6, perhaps resulted from the dominated regeneration process caused by high concentration of H+ and the strong electrostatic repulsion[56]. And the value of IER is extremely close to 1 here, indicating that the phenomenological ion exchange domains, the fundamental mechanism might be co-precipitaion/complexation with high bonding strength and cation-π interaction which is irrespective of pH change.

The value of Δqoc and the adsorbed amount of Pb(Ⅱ) increase with initial pH, but the IER decreases, which can be interpreted by the weakening electrostatic repulsion and the preferentially enlarged exposure of aromatic C=C/C=N to Pb(Ⅱ) which are potential adsorptive sites but non-essentially related to the phenomenological ion exchange.

  • Sharma, G., D. Pathania, and M. Naushad," Preparation, characterization, and ion exchange behavior of nanocomposite polyaniline zirconium(IV) selenotungstophosphate for the separation of toxic metal ions", IONICS 21, No. 4, 2015, pp. 1045-1055.

Please check in Line471-481 in the revised manuscript.

  1. What functional groups are present on surface of material which are responsible for adsorption.

Answer: The DRIFT and XPS results indicate that O-containing groups(including C-O-C,C-O and C=O) ,N-containing functional groups(including pyrrolic groups, -NH-, -NH2 and N=O) and aromatic groups play important part in the adsorption of Pb(Ⅱ) by AHC0.1. Specifically, as depicted in Fig13, C-O and C=O can adsorbe Pb2+ by complexation or coordination; C-O-C, -NH-, -NH2 and N=O can coordinate with Pb2+, and aromatic groups and pyrrolic groups can adsorbe Pb2+ through cation-π interaction.

Please check in Line 497-502 in the revised manuscript.

  1. In conclusion section state why, your material is better adsorbent.

Answer: The conclusion has been improved as follow: A cost-effective adsorbent for Pb(Ⅱ) was prepared by hydrothermal carbonization using sewage sludge and inexpensive citric acid as feedstock. The introduction of citric acid into the preparation of sewage sludge-derived hydrochar can effectively develop its pore structure, increase its surface functional groups and improve its thermal and chemical stability. Demonstrated by the result of adsorption experiments, AHC0.1 has great adsorption ability for its fast kinetics and maximum adsorption capacity of 60.88 mg·g-1 which is competitive to other adsorbents. AHC0.1 adsorbs and removes Pb(Ⅱ) mainly through its phenomenological ion exchange with other metal cations especially Ca2+ and Mg2+. And the adsorption can be mainly attributed to co-precipitation, complexation, cation-π interaction and electrostatic interaction. Due to the low-cost feedstock, the facile production and the significant potential of the produced hydrochar as adsorbents for heavy metal removal, the joint hydrothermal carbonization of citric acid and sewage sludge can achieve the value-added utilization of sewage sludge as adsorbents.

Please check in Line 505-516 in the revised manuscript.

  1. Compare your adsorbent results with other reported work.

Answer: The comparison has been given as Table7.

Although the specific surface area of AHC0.1 is relatively low(59.95m2·g-1), but its maximum adsorption capacity of Pb() (60.88 mg·g-1) is superior or at least comparable to that of the majority of sewage sludge drived hydrochars, sewage sludge biochars and some hydrochars derived from other biomass(listed in Table7), which might benefit from the rich O-containing/N-containing functional groups and inorganics(silicates and phosphates) in AHC0.1.

Table7. Comparison of maximum adsorption capacity of Pb(Ⅱ) onto various adsorbents

Adsorbents

Qmax  (mg·g-1)

SBET-N2 (m2·g-1)

Reference

H2O2 modified peanut hull hydrochar

22.82

1.4

[50]

SiO2/polysaccharide-hydrochar composite

52

21.76

[51]

LDH/sewage sludge-hydrochar composite

62.44

-

[52]

KOH activated sewage sludge biochar

57.48

907.95

[53]

Sewage sludge-based biochar

18.20

23.70

[54]

Sewage sludge-based biochar

51.20

-

[55]

SSHC

44.34

45.53

This study

AHC0.1

60.88

59.95

This study

[50] Xue, Y., B. Gao, Y. Yao, M. Inyang, M. Zhang, A.R. Zimmerman, and K.S. Ro," Hydrogen peroxide modification enhances the ability of biochar (hydrochar) produced from hydrothermal carbonization of peanut hull to remove aqueous heavy metals: Batch and column tests", CHEM ENG J Vol. 200-202, 2012, pp. 673-680.

[51] Li, Y., K. Li, M. Su, Y. Ren, Y. Li, J. Chen, and L. Li," Fabrication of carbon/SiO2 composites from the hydrothermal carbonization process of polysaccharide and their adsorption performance", CARBOHYD POLYM Vol. 153, 2016, pp. 320-328.

[52] Luo, X., Z. Huang, J. Lin, X. Li, J. Qiu, J. Liu, and X. Mao," Hydrothermal carbonization of sewage sludge and in-situ preparation of hydrochar/MgAl-layered double hydroxides composites for adsorption of Pb(II)", J CLEAN PROD Vol. 258, 2020, pp. 120991.

[53] Zhang, J., J. Shao, Q. Jin, Z. Li, X. Zhang, Y. Chen, S. Zhang, and H. Chen," Sludge-based biochar activation to enhance Pb(II) adsorption", FUEL Vol. 252, 2019, pp. 101-108.

[54] Zhang, W., S. Mao, H. Chen, L. Huang, and R. Qiu," Pb(II) and Cr(VI) sorption by biochars pyrolyzed from the municipal wastewater sludge under different heating conditions", BIORESOURCE TECHNOL Vol. 147, 2013, pp. 545-552.

[55] Ho, S., Y. Chen, Z. Yang, D. Nagarajan, J. Chang, and N. Ren," High-efficiency removal of lead from wastewater by biochar derived from anaerobic digestion sludge", BIORESOURCE TECHNOL Vol. 246, 2017, pp. 142-149.

Please check in Line 442-449 in the revised manuscript.

  1. Improve the quality of figures specifically XRD.

Answer: The quality of figures have been improved, especially XRD.

Please check in Line 338-340 in the revised manuscript.

Fig. 5 XRD patterns of SS-derived hydrochars and Pb@AHC0.1

  • If possible, provide pictorial presentation showing mechanism about adsorption.

Answer: The schematics has been given as Fig.13.

Please check in Line484-486 in the revised manuscript.

Fig.13 Schematic of Pb(Ⅱ) adsorption mechanism by AHC0.1

Round 2

Reviewer 2 Report

Paper can be accepted.